# Deep-sequencing of viral genomes from a large and diverse cohort of treatment-naive HIV-infected persons shows associations between intrahost genetic diversity and viral load

Migle Gabrielaite[1⊙]*, Marc Bennedbæk[2⊙]*, Malthe Sebro Rasmussen[1,3], Virginia Kan[4], Hansjakob Furrer[5], Robert Flisiak[6], Marcelo Losso[7], Jens D. Lundgren[2], INSIGHT START Study Group, Rasmus L. Marvig[1]

1 Center for Genomic Medicine, Rigshospitalet, Copenhagen, Denmark, 2 Centre of Excellence for Health, Immunity and Infections, Department of Infectious Diseases, Rigshospitalet, University of Copenhagen, Copenhagen, Denmark, 3 Section of Computational and RNA Biology, Department of Biology, University of Copenhagen, Copenhagen, Denmark, 4 Veterans Affairs Medical Center and The George Washington University School of Medicine and Health Sciences, Washington, District of Columbia, United States of America, 5 Department of Infectious Diseases, Bern University Hospital, University of Bern, Bern, Switzerland, 6 Department of Infectious Diseases and Hepatology, Medical University of Bialystok, Bialystok, Poland, 7 Hospital General De Agudos J M Ramos Mejía, Buenos Aires, Argentina

⊙ These authors contributed equally to this work.
* migle.gabrielaite@gmail.com (MG); mcbk@ssi.dk (MB)

**Data Availability Statement:** All non-personally identifiable data, e.g. all output from Virvarseq, and relevant code is deposited at Dryad repository:

## Abstract

### Background

Infection with human immunodeficiency virus type 1 (HIV) typically results from transmission of a small and genetically uniform viral population. Following transmission, the virus population becomes more diverse because of recombination and acquired mutations through genetic drift and selection. Viral intrahost genetic diversity remains a major obstacle to the cure of HIV; however, the association between intrahost diversity and disease progression markers has not been investigated in large and diverse cohorts for which the majority of the genome has been deep-sequenced. Viral load (VL) is a key progression marker and understanding of its relationship to viral intrahost genetic diversity could help design future strategies for HIV monitoring and treatment.

### Methods

We analysed deep-sequenced viral genomes from 2,650 treatment-naive HIV-infected persons to measure the intrahost genetic diversity of 2,447 genomic codon positions as calculated by Shannon entropy. We tested for associations between VL and amino acid (AA) entropy accounting for sex, age, race, duration of infection, and HIV population structure.

https://doi.org/10.5061/dryad.6djh9w13s. Personal identifiable data can be accessed through agreement with the INSIGHT START Study Group (http://www.insight-trials.org/research_proposal) to respect the donor consent of the participants in the START study. Personal identifiable data include individual-level metadata.

**Funding:** The work was supported by the Danish National Research Foundation (grant 126). The funders had no role in study design, data collection and analysis, decision to publish, or preparation of the manuscript.

**Competing interests:** The authors have declared that no competing interests exist.

## Results

We confirmed that the intrahost genetic diversity is highest in the *env* gene. Furthermore, we showed that mean Shannon entropy is significantly associated with VL, especially in infections of >24 months duration. We identified 16 significant associations between VL (p-value<$2.0 \times 10^{-5}$) and Shannon entropy at AA positions which in our association analysis explained 13% of the variance in VL. Finally, equivalent analysis based on variation in HIV consensus sequences explained only 2% of VL variance.

## Conclusions

Our results elucidate that viral intrahost genetic diversity is associated with VL and could be used as a better disease progression marker than HIV consensus sequence variants, especially in infections of longer duration. We emphasize that viral intrahost diversity should be considered when studying viral genomes and infection outcomes.

## Trial registration

Samples included in this study were derived from participants who consented in the clinical trial, START (NCT00867048) (23), run by the International Network for Strategic Initiatives in Global HIV Trials (INSIGHT). All the participant sites are listed here: http://www.insight-trials.org/start/my_phpscript/participating.php?by=site

## Author summary

Viral intrahost genetic diversity complicates the cure of HIV; nevertheless, there is a lack of large and diverse cohort studies based on near-full and deep-sequenced HIV genomes. Here, we analysed deep-sequenced viral genomes from 2,650 demographically diverse and treatment-naive HIV-infected persons to measure the intrahost genetic diversity of 2,447 genomic codon positions as calculated by Shannon entropy. First, we verified that intrahost genetic diversity is highest in the env gene. Then, we identified that mean Shannon entropy positively associates with viral load, mostly in infections of >24 months duration. Finally, we showed that intrahost diversity in 16 HIV genomic positions significantly associated with viral load and explained 13% of the variance in viral load, whereas equivalent analysis based on variation in HIV consensus sequences explained only 2% of viral load variance. Overall, our study shows that higher viral intrahost genetic diversity positively associates with viral load and is a better disease progression marker than variation of HIV consensus sequences. Our findings further suggest that better understanding of pathogen genomics is required to better address infectious diseases.

## Introduction

Human immunodeficiency virus type 1 (HIV) shows high genetic diversity between and within the human hosts [1,2]. While within-host viral diversity is initially limited by the transmission bottleneck, HIV genomes quickly diversify as a result of a high viral replication rate and high error rates during replication [3]. Such high genetic diversity allows the virus to avoid the host's immune response and can lead to development of drug resistance during treatment [1]. HIV-1 set-point viral load (spVL), the approximately stable viral load (VL) measured

by plasma HIV-RNA in the early chronic phase of HIV-infection, is associated with clinical progression of HIV-infection [4]. SpVL has been shown to be associated with human genetics and, even more pronounced, with viral genetics [5].

Intrahost viral genetic diversity balances between adaptation to the host environment and maintenance of replication ability, and it allows to explore intrahost HIV evolution in cross-sectional datasets. Even though longitudinally collected datasets are preferred for intrahost HIV evolution, cross-sectional datasets are easier to collect and scale up [6,7]. Among various methods for intrahost genetic diversity quantification, Shannon entropy is commonly used and easy to interpret [8–10]. Shannon entropy reflects both the number of present variants and their frequencies [9]. Shannon entropy is highest at the genomic position when all possible encoded amino acids (AAs) are present at equal frequencies [11]. Higher HIV intrahost diversity could be a result of the faster evolution of HIV escape mutants or, alternatively, could be a result of a more robust host immune response to HIV [12].

To date, the relationship between intrahost genetic diversity and disease progression markers, e.g. VL, have been investigated in relatively highly selected and small cohorts (up to 187 individuals in Hightower et al., 2012 [10]) and have typically been based on techniques that allows for only low depth sampling for variation in a minor part of the HIV genome [10,12–21]. Investigations based on near-full and deep genome sequencing of HIV genomes from large and diverse cohorts are required to further elucidate the relationship between intrahost genetic diversity and disease progression and generalize previous findings from smaller and more homogenous cohorts.

Participants from across 35 countries in the Strategic Timing of AntiRetroviral Treatment (START) clinical trial represent a demographically diverse cohort of treatment naïve HIV-positive persons infected with a range of different HIV subtypes and for which near-full viral genomes have been deep-sequenced [22–25]. Hence, by using genomic data from the START cohort, we aimed to identify how Shannon entropy at AA positions as a measure of intrahost diversity varies across the HIV genome, and how it relates to the VL. We further aimed to identify the AA positions in which Shannon entropy was most associated with the VL, and finally, we compared whether interhost or intrahost HIV genetic variation better explained variance in VL.

## Methods

### Ethics statement

Samples included in this study were derived from participants who consented in the clinical trial, START (NCT00867048) [23], run by the International Network for Strategic Initiatives in Global HIV Trials (INSIGHT). The study was approved by the institutional review board or ethics committee at each contributing center, and written informed consent was obtained from all participants. All informed consents were reviewed and approved by participant site ethics review committees. All the participant sites are listed here: http://www.insight-trials.org/start/my_phpscript/participating.php?by=site.

### Study population

Samples included in this study were derived from participants from the START trial, which included 4,685 participants fulfilling the following criteria: 1) a CD4$^+$ cell count >500 cells/μl at baseline, 2) no history of AIDS, 3) no previous history of antiretroviral treatment and 4) were above 18-years-old at study entry [23,26]. For this study, we included participants which had baseline plasma samples with a VL measurement ≥ 1,000 copies/mL and viral genomes sequenced by next-generation sequencing (N = 3,785).

### HIV sequencing, alignment, variant calling, and subtyping

The detailed laboratory procedure is described in Bennedbæk *et al.*, 2021 [25]. In short, viral RNA was extracted from plasma and amplified using two amplicons spanning 7,125 nt (HIV-1 HXB2 genome regions 1,485–5,058 and 5,967–9,517) of the 9,719 nt (74%) HIV genome and sequenced by next-generation sequencing (S1 Fig). Reads were first aligned against the GRXh37.p13 human reference genome with Bowtie2 v2.2.8 and only read pairs with neither read aligned to the human genome were retained. Virvarseq [27] was used for alignment of cleaned reads and AA variant calling with the HXB2 HIV reference genome (GenBank accession number K03455.1) with default parameters. Virvarseq alignment, realignment, and AA variant calling were performed per gene basis (*rev* and *tat* genes were split into separate analyses for each reading frame). The two amplicons with primer regions excluded did not completely cover the genes *gag*, *pol*, *rev*, and *tat*, therefore, variant calls for the gene coding for the Gag protein were considered from HXB2 reference genome codon position 240, for Pol until AA position 980, for Rev from position 7, and for Tat from position 52. No sequencing was available for Vpr and Vif (S1 Fig). Minimum Phred quality scores for all called codons were >20. Variant calling was only performed for samples that had at least 10% of the gene covered by aligned reads with a minimum 10-fold sequencing depth. Variants were called if there was a minimum of 500-fold sequencing depth and a minimum of 1% of aligned reads supported the variant. Samples which had ≤500 codon positions with ≥500-fold sequencing depth were excluded from the analysis (N = 1,135). Consensus variants were called if there was a minimum of 10-fold sequencing depth and minimum 50% of aligned reads supported the variant.

Additional analyses were performed to test the robustness of the results with minimum sequencing depth thresholds set to 300-fold and 1,000-fold, respectively (instead of a 500-fold threshold). In these analysis, 1,002 and 1,379 samples with ≤500 codon positions with ≥500-fold sequencing depth were excluded from the analysis, respectively (instead of 1,135 samples).

The detailed subtyping description is available in other publications [22,25]. In short, the consensus sequences were used for subtyping using REGA version 3.0 [28] and the subtypes were confirmed by a manual inspection. Samples with more than three predicted subtypes were labelled as "Mixed". If less than 100 participants were assigned a particular subtype, the subtype was grouped as "Other" in this study.

### Shannon entropy

Shannon entropy was used as a measure of intrahost diversity and was calculated for each AA position in each sample. Shannon entropy $H_i$ at the position $i$, where

$$H_i = -\sum_a f_{a,i} log_2 f_{a,i},$$

and $f_{a,i}$ is the frequency of the amino acid (AA) $a$ at position $i$. Mean Shannon entropy at the AA position was used for identification of the most variable HIV-1 genomic positions. The threshold defining the most variable positions was arbitrarily chosen to be 0.3. Sample's mean Shannon entropy was used for correlation with $log_{10}(VL)$ and regression analyses.

### Statistical analysis

The association analysis was performed with R [29]. A multiple linear regression model was fitted to assess associations of VL with sample's mean Shannon entropy. VL measurements were $log_{10}$ transformed prior to analysis. Age, sex assigned at birth, self-reported race, duration

of infection, the first four principal components of the HIV genetic data, subtype, and the average sequencing depth of positions with ≥500-fold sequencing depth (hereafter referred as average sequencing depth) were considered as covariates to account for population structures. Bonferroni correction was used to account for multiple testing in the association between VL and Shannon entropy at AA positions.

Principal component analysis (PCA) was performed using the consensus AA calls with EIGENSOFT software v6.1.4 [30]. Furthermore, only AA variants which had allele frequency ≥5% and ≤95% were used. PCA analysis was based on 2,180 AA variants and the first four principal components (PCs) were used for regression model construction to account for the population structure as previously described [24]. R software [29] was used for explained variance analysis and visualizations. The explained variance was calculated by comparing the variable's sum of squares with the total sum of squares from analysis of variance (ANOVA) analysis. The final model during the model selection was chosen based on the ANOVA analysis when comparing to previous models. The significant associations from the multiple linear association analysis for the Shannon entropy of the 2,447 HIV AA positions using the same covariates as in the previous regression analysis were referenced against the best-defined cytotoxic T lymphocyte (CTL) epitopes from the HIV molecular immunology database [31]. Additional robustness re-analyses were performed using 300-fold and 1000-fold sequencing depth thresholds, resulting in 2,542 and 2,085 HIV AA positions being considered, respectively.

A 5-fold cross validation was performed by randomly splitting the dataset into 5 equal parts, using one part as a test dataset and the rest as training data to calculate parameters for the regression model. The model was evaluated by calculating root-mean-square deviation (RMSE) and R2 parameters after fitting the model on the test and train datasets.

The duration of infection variable was based on a study by Sharma *et al.* 2019 [32] where the duration of infection was determined to be ≤6 months based on multi assay algorithm (MAA) serological and non-serological markers of recent infection or the self-reported date of infection. 6–24 month duration of infection was determined 1) if MAA failed to confirm a recent infection, infection date was unknown and the diagnosis date was <6 months before randomization or 2) if HIV diagnosis was 6–24 months before randomization. Participants were considered to be infected for >24 months if their HIV diagnosis was >24 months before randomization.

## Dryad DOI

https://doi.org/10.5061/dryad.6djh9w13s [26]

# Results

## Study participants and genome sequencing

A total of 3,785 START clinical trial participants had HIV genomes sequenced [22,24,25]. After read alignment to the HXB2 reference genome, 1,135 genomes had 500 or less (out of 2,549) codon positions covered by ≥500 reads. These genomes were removed from further analysis leaving 2,650 HIV genomes in the study (S1 Table). It should be noted that the samples of the excluded genomes had a significantly lower VL than the included HIV genomes (Wilcoxon signed-rank test, p-value<2.2x10$^{-16}$). The included genomes on average had a 5,521-fold (range 214–40,501-fold) sequencing depth, and an average sequencing depth of 11,638-fold (range 667–163,667-fold) for the positions included in this analysis (i.e., the positions with ≥500-fold sequencing depth). VL of the included samples correlated weakly with sequencing depth (Pearson's correlation coefficient– 0.12).

## Shannon entropy calculation

While we calculated Shannon entropy for all 2,549 HIV genomic codon positions within the targeted genomic regions (Fig 1), 102 positions, which had sufficient sequencing depth in ≤800 samples, were excluded from further analysis as these positions showed an aberrant number of samples with sufficient sequencing depth (may be a result of unsuccessful mapping of reads to highly variable genomic regions) and coincided with high entropy. The excluded positions were all either within Env or near the start or end of amplicons (Fig 1 and S2 Table). Accordingly, a total of 2,447 positions were included with a distribution over the HIV genome as follows: Gag—262 positions, Pol—979 positions, Env—799 positions, Vpu—78 positions, Tat—47 positions, Rev—107 positions, Nef—207 positions.

## The most variable positions in the sampled HIV genomes

A total of 2,447 AA positions were considered for the most variable position identification. Overall, 36 positions had mean Shannon entropy ≥0.3 (S3 Table and Fig 1): five high entropy positions were in Gag, two in Pol, and 29 in Env. Mean Shannon entropy was significantly higher (Wilcoxon signed-rank test, p-value<$2.2 \times 10^{-16}$) within Env variable loop regions than in other Env positions. The entropy distributions in the most variable positions were similar between different races and subtypes (S1 Table and S2 and S3 Figs).

If we performed the analysis using a more stringent 1,000-fold requirement for sequencing depth of included positions (instead of the 500-fold requirement in main analysis), we identified the same positions to show high entropy; except if the positions were excluded from the analysis due to the increased requirement for sequencing depth (S4 Table). If we used a less stringent 300-fold requirement for sequencing depth, we also identified the same positions to show high entropy; yet we then also identified additional 22 high entropy positions that did not have enough sequencing depths to be included in the more stringent analyses (S4 Table).

## Intrahost diversity correlation with viral load

To investigate the linear dependence between sample's mean Shannon entropy and $\log_{10}$ VL, and the duration of infection, respectively, we performed a univariate linear regression of $\log_{10}$ VL and the sample's mean Shannon entropy, and explored their differences based on the duration of infection (Fig 2). We observed that entropy increased with VL regardless of the duration of infection. Sample's mean Shannon entropy alone, as deduced by ANOVA analysis, explained 4.14% of the observed VL variance. Contrarily, a univariate model of $\log_{10}$ VL association with the average sequencing depth explained 0.68% of the variance in VL (S5 Table).

The median VL was lowest in the infections with >24 months duration while the median of the sample's mean Shannon entropy was highest in this group. Pearson's correlation coefficient between VL and the sample's mean Shannon entropy was 0.20, 0.19 and 0.33 for infection duration of <6 months, 6–24 months and >24 months, respectively. Sample's mean Shannon entropy, as deduced by ANOVA analysis, explained 3.91%, 3.56%, and 10.70% of the variance in VL for infection duration of <6 months, 6–24 months and >24 months, respectively. The explained variance using a 300-fold and 1000-fold sequencing depth thresholds, respectively, was comparable to the main analysis (S4 Table). ANOVA analysis revealed that all three variables: sample's mean Shannon entropy (p-value<$2.2 \times 10^{-16}$), duration of infection (p-value = $3.5 \times 10^{-9}$) and their interaction (p-value = $2.0 \times 10^{-2}$) have a significant effect on VL.

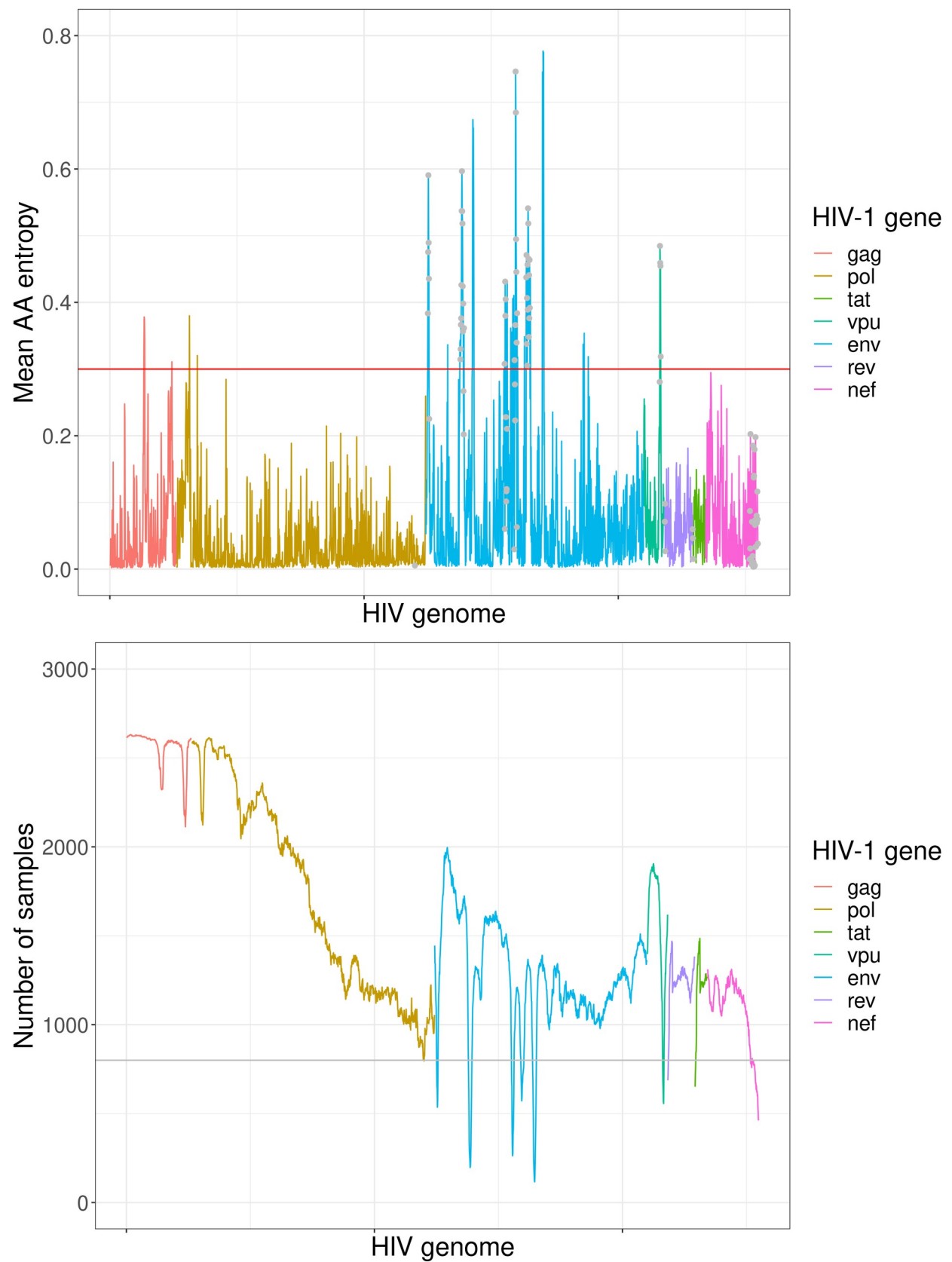

**Fig 1.** Mean Shannon entropy over the HIV genome (top) and number of samples with minimum 500-fold genome sequencing depth over the genome (bottom). A total of 2,447 HIV AA positions are shown on the x-axis. Gag positions 1–239, Pol positions 981–1004, Rev positions 1–6 and Tat positions 1–51 are not shown, as they were not covered by the amplicons. Red horizontal line in the top figure shows the used Shannon entropy threshold. Grey points at the top plot mark the HIV genome positions which were excluded from the analysis (≤800 samples). Grey horizontal line at the bottom plot shows the threshold for the minimum number of samples.

## Mean intrahost diversity association with viral load

Next, we built a multiple linear regression model to further explore the relationship between $\log_{10}$ VL and Shannon entropy (S5 Table). The final model included participant's age, sex assigned at birth, race, duration of infection and the first four PCs (which are used to account for the viral genetic population structure) as covariates (S5 Table). In total, the final regression model (i.e., sample's mean Shannon entropy together with all the covariates) explained 10.3% of the variance in VL. The explained variance in VL for the re-analysis using a 300-fold and 1000-fold sequencing depth thresholds, respectively, was comparable to the 500-fold threshold used in the main analysis (S4 Table). The final model including only the participants infected for >24 months explained 13.0% of the variance in VL. If we added the average sequencing depth as a covariate to the multiple linear regression model together with other covariates, the model explained 10.7% rather than 10.3% of the variance (S5 Table). To confirm that our findings are not affected by overfitting, we performed a 5-fold cross validation analysis. The average RMSE of the training and testing dataset was 0.574 (0.571–0.577) and 0.578 (0.565–0.587), respectively. The explained variance was 9.3% for the test dataset and 10.4% for the train dataset. Shannon entropy averaged per gene for each sample in the final multiple linear regression model (Model 6 in S5 Table) did not associate significantly with VL (p-value = 0.31 for Gag, p-value = 0.24 for Pol, p-value = 0.48 for Env, p-value = 0.48 for Vpu, p-value = 0.33 for Rev, p-value = 0.34 for Tat, p-value = 0.21 for Nef).

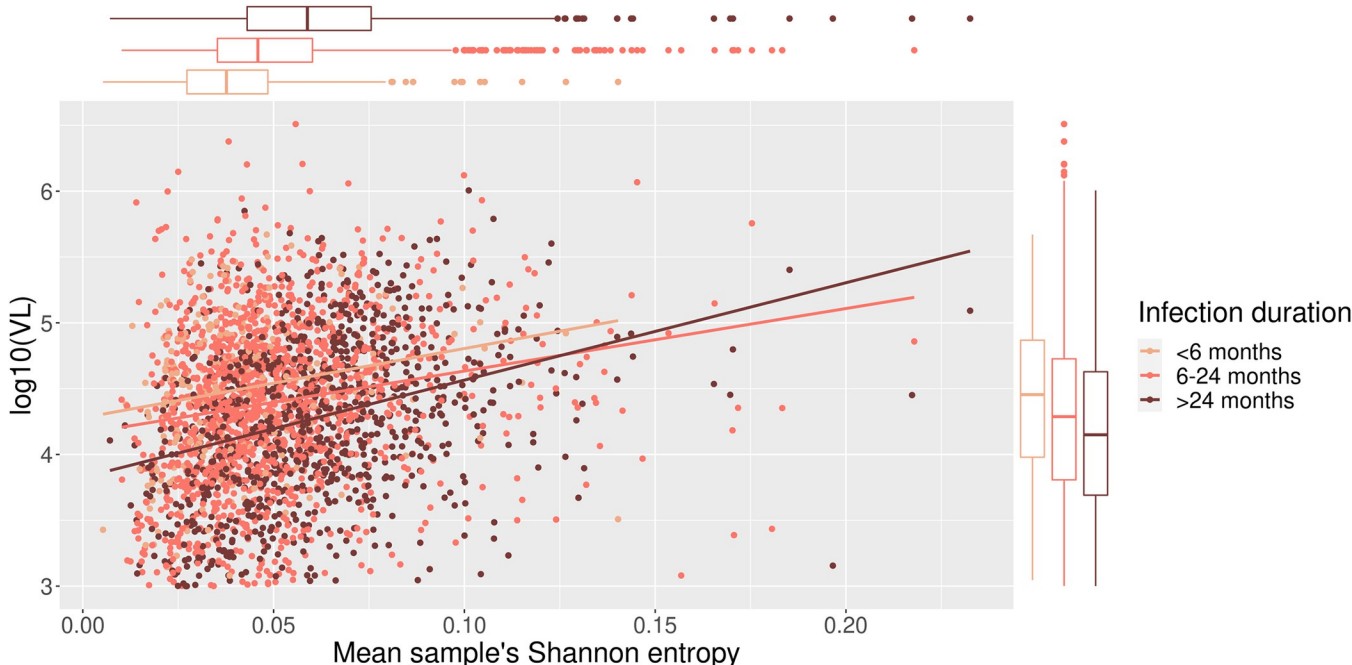

**Fig 2. The relationship between sample's mean Shannon entropy, viral load (VL) and infection duration.**

### HIV intrahost position entropy association with viral load

To disentangle the effect intrahost diversity at the specific HIV genome positions has on $\log_{10}$ VL, we performed the multiple linear association analysis for the Shannon entropy of 2,447 HIV AA positions using the same covariates as in the previous regression analysis (i.e., participant's age, sex assigned at birth, race, duration of infection and the first four PCs). In 16 cases (six positions in Pol, nine positions in Env and one position in Nef; Fig 3) the VL association with the Shannon entropy at the specific HIV AA position was below Bonferroni-corrected p-value threshold (p-value$<2.0 \times 10^{-5}$). In all 16 positions the association was driven by a general trend and not a few outliers (Table 1 and S4 Fig). Together Shannon entropy values from the 16 associated AA positions explained 12.6% of the variance in VL. The full model explained 21.9% of the variance in the VL (S6 Table). Average RMSE values from train and test data in the 5-fold cross validation analysis were similar (S7 Table). The joint model including all 16 positions revealed that only half of the variants were significantly associated with VL when considered together (S5 Table) which is likely a result of multicollinearity between the Shannon entropy in different HIV genomic positions (S5 Fig). The regression model with the consensus AA variants using the same covariates showed that 13 significantly associated consensus AA variants explain 0.01−0.65% of the variance in VL and the full model with the covariates explained 6.2% of the variance in VL (S8 Table).

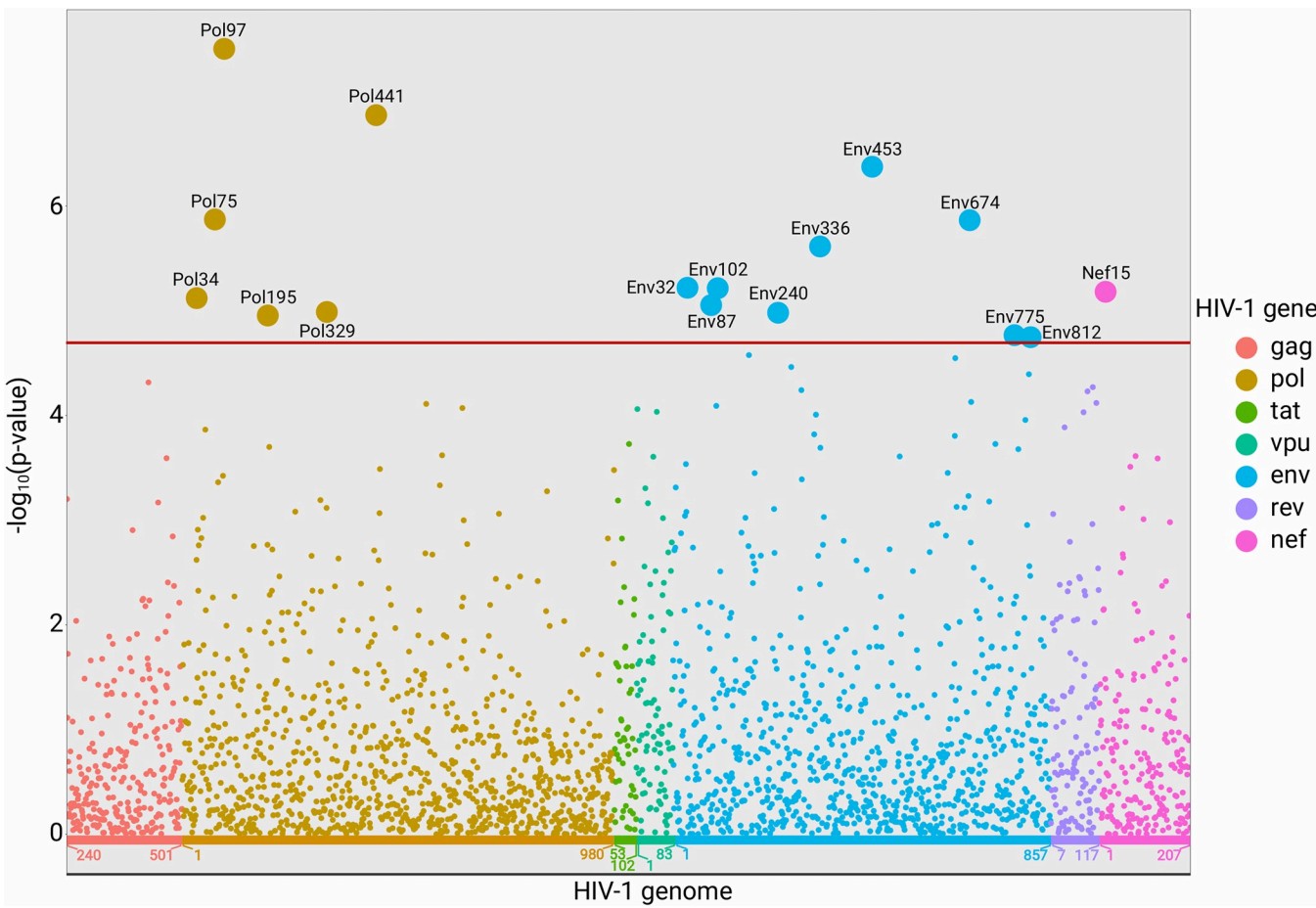

**Fig 3. Manhattan plot of the associations between viral load (VL) and Shannon entropy at AA positions projected on the HIV genome.** Red horizontal line marks the Bonferroni p-value threshold (p = $2.0 \times 10^{-5}$). The x-axis denotes gene AA positions. A total of 2,447 HIV AA positions are shown on the x-axis. Gag positions 1–239, Pol positions 981–1004, Rev positions 1–6 and Tat positions 1–51 are not shown, as they were not covered by the amplicons.

**Table 1. Sixteen HIV amino acid (AA) positions where Shannon entropy was significantly associated with viral load (VL).**

| Gene | AA position | Effect size (95% CI) | p-value | Mean Shannon entropy | CTL epitope coordinates (HLA alleles) [a] |
|------|-------------|----------------------|---------|----------------------|--------------------------------------------|
| Pol | 34 | 0.229 (0.129–0.330) | $7.53 \times 10^{-6}$ | 0.10 | |
| Pol | 75 | 0.262 (0.156–0.369) | $1.32 \times 10^{-6}$ | 0.07 | |
| Pol | 97 | 0.221 (0.143–0.298) | $3.15 \times 10^{-8}$ | 0.19 | 90–98 (B*44) |
| Pol | 195 | 0.161 (0.089–0.233) | $1.09 \times 10^{-5}$ | 0.28 | 188–198 (C*01:02) |
| Pol | 329 | 0.263 (0.146–0.380) | $1.02 \times 10^{-5}$ | 0.07 | 328–336 (A*30:02, C*12:02) |
| Pol | 441 | 0.338 (0.213–0.463) | $1.33 \times 10^{-7}$ | 0.09 | |
| Env | 32 | 0.168 (0.095–0.240) | $5.96 \times 10^{-6}$ | 0.22 | 31–39 (B*18:01,B*44); 31–40 (B*44:02) |
| Env | 87 | 0.136 (0.076–0.196) | $8.71 \times 10^{-6}$ | 0.34 | |
| Env | 102 | 0.340 (0.193–0.487) | $6.03 \times 10^{-6}$ | 0.06 | |
| Env | 240 | 0.162 (0.090–0.234) | $1.03 \times 10^{-5}$ | 0.23 | |
| Env | 336 | 0.168 (0.098–0.237) | $2.41 \times 10^{-6}$ | 0.34 | |
| Env | 453 | 0.400 (0.246–0.555) | $4.17 \times 10^{-7}$ | 0.07 | |
| Env | 674 | 0.246 (0.147–0.345) | $1.36 \times 10^{-6}$ | 0.18 | |
| Env | 775 | 0.353 (0.192–0.513) | $1.70 \times 10^{-5}$ | 0.07 | 770–780 (A*03:01, A*30:01) |
| Env | 812 | 0.274 (0.149–0.399) | $1.78 \times 10^{-5}$ | 0.10 | 805–814 (B*40:01) |
| Nef | 15 | 0.210 (0.119–0.301) | $6.50 \times 10^{-6}$ | 0.22 | 13–20 (B*08:01) |

[a] CTL epitope reference: https://www.hiv.lanl.gov/content/immunology/tables/optimal_ctl_summary.html

Only three associated AA positions (Pol34, Env453 and Env812) had no mutation entries based on the data from HIV mutation browser [33], i.e., were conserved positions. Seven out of 16 associated AA positions overlapped with one or more best-defined CTL epitopes (Table 1). We did not find that the associated AA positions overlapped significantly differently with the CTL epitope positions than expected by chance (Fisher's exact test; p-value = 1) [31].

## Discussion

While we and others have associated variation in consensus sequences of the HIV-1 quasispecies with disease progression markers in studies with >1,000 individuals [5,24,34], equivalent analyses of intrahost viral diversity have, despite showing strong associations, been conducted on cohorts with only up to 187 individuals [10] and have typically been based on low depth sampling of genetic variation in only a minor part of the HIV genome [10,12–21]. Consequently, our study addresses the need for associations between intrahost diversity and disease progression markers in a large and demographically diverse cohort with near-full deep-sequenced HIV genomes. Moreover, we compare the strength of the association between VL and intrahost diversity with VL and consensus HIV AA variants (Table 2).

We first identified the most variable positions of the HIV genomes. Then, we built a multiple regression model to associate the differences in VL with the sample's mean intrahost diversity, and finally, we identified 16 HIV genome positions where intrahost diversity of the position was associated with VL, which is an established marker of HIV disease progression [4]. Note, while VL is a useful measure, VL tends to increase over time and to different degrees in different individuals, and this increase may be an important component of progression [35].

Our most variable HIV genome position analysis confirmed that intrahost diversity is not equally distributed across the HIV genome and is higher in Env with several higher entropy positions in Gag and Pol [36,37]. Additionally, we could not find an association of HIV intrahost diversity and the HIV subtype or the race of the participant. It is important to emphasize

**Table 2. Variance in viral load (VL) explained by linear regression analysis with different models.** PC1-PC4 denote first four principal components of principal component analysis performed on consensus AA variants to account for the viral population structure. The 16 and 13 HIV AA positions are all positions significantly associated with VL when position-wise Shannon entropy or consensus variants, respectively, are used to explain VL.

| Model | % of VL variance explained |
|---|---|
| VL ~ Sample's mean Shannon entropy (all samples) | 4.14 |
| VL ~ Sample's mean Shannon entropy (samples from <6 months infection duration) | 3.91 |
| VL ~ Sample's mean Shannon entropy (samples from 6–24 months infection duration) | 3.56 |
| VL ~ Sample's mean Shannon entropy (samples from >24 months infection duration) | 10.7 |
| VL ~ Average sequencing depth | 0.68 |
| VL ~ Sample's mean Shannon entropy + age + sex + race + duration of infection + [PC1-PC4] | 10.3 |
| VL ~ Sample's mean Shannon entropy + age + sex + race + duration of infection + [PC1-PC4] + average sequencing depth | 10.7 |
| VL ~ Shannon entropy at 16 HIV AA positions + age + sex + race + duration of infection + [PC1-PC4] | 21.9 |
| VL ~ Consensus variants at 13 HIV AA positions + age + sex + race + duration of infection + [PC1-PC4] | 6.2 |

that we identified the most variable HIV genome positions as an absolute measure across the genome and not relative to the background variability of a given genomic region, i.e., we used the same threshold for genome variability throughout the genome without considering the overall higher or lower variability of the specific gene. Furthermore, our analysis of Shannon entropy averaged per gene for each sample showed that that the relationship between Shannon entropy and VL is not driven by intrahost diversity in one particular gene.

The relationship between VL and sample's mean Shannon entropy revealed a positive correlation between these parameters where the infections of short duration showed a weak correlation and infections of longer duration were moderately correlated with VL which could be explained by the initial transmission bottleneck followed by viral diversification [38–40]. It is in agreement with the established knowledge that spVL as a prognostic marker can only be employed in chronic infection [41]. These findings concur with a study by Kafandro *et al.* 2017, where the authors showed that Shannon entropy can confidently discriminate between recent and chronic infections [42]. Moreover, the lower VL in chronic infections than in recent infections might be due to the nature of the cohort where participants with a longer infection duration who still had a high CD4+ cell count are likely to better control the infection innately [7]. Finally, our results indicate that the sample's mean Shannon entropy, especially in infections of longer duration, is informative when explaining VL variation among patients since it together with other covariates in our model explains >10% of the overall variance in VL. Our findings are in line with a smaller study by Bello *et al.* 2007 which found a significant positive correlation between viral genetic changes and VL in long-term nonprogressors (LTNP) [15] even though our study was not limited to LTNP. Note, that the observed results could not be explained by the fact that more variability is observed in higher sequencing depth. Additionally, while the directionality of the relationship between VL and intrahost diversity cannot be determined from our study, the work by Bello *et al.* 2007 showed that increase in intrahost diversity directly leads to the increase of VL [15].

The regression model in our main analysis did not include HIV subtypes as subtype assignment did not improve the model significantly; instead, the first four principal components of genome consensus sequences were sufficient to account for the genomic population structure. We also tested how robust our results were to changes in the sequencing depth required to

include positions (and thus samples) and to the addition of the average sequencing depth as a covariate to the model. Results were robust to both tests, and, while VL of the included samples correlated weakly with sequencing depth, we found average sequencing depth alone to explain little of the variance in VL. Finally, we obtained similar results in our analysis when we applied 5-fold cross validation, suggesting that the model was not overfitted.

Our analysis showed that VL was significantly associated with the intrahost diversity of 16 HIV AA positions and as shown by the joint multiple regression model, the intrahost diversity in these HIV AA positions is not independent of each other and has a collinear relationship. One explanation for VL association with Shannon entropy at these HIV AA positions could be that these positions are less conserved and acquire mutations faster. We hypothesize that these processes correlate with the acquisition of other adaptive mutations, and, therefore, higher VL [6]. On the other hand, higher Shannon entropy in the associated positions could provide an advantage to the virus when avoiding the immune response, so the higher VL would be a direct outcome of the increased entropy as some of the associated positions are a part of best-defined CTL epitopes. [31,36,37]. While both explanations are plausible, we did not find support that increased Shannon entropy at these positions could give rise to a direct increase in VL. Furthermore, by using a multiple regression model with HIV consensus sequences and the same covariates as in this work, we confirmed that VL associates poorly with HIV consensus sequences [21]. Therefore, we conclude that the HIV genome position entropy is a substantially better marker of VL than the HIV mutations identified from consensus sequences [24,34]. However, the association of VL with Shannon entropy in the 16 identified HIV AA positions should be further tested in other studies, and more work is needed to define how intrahost genome diversity at an early point in time could be a predictor of later changes in CD4+ cell count, VL or other markers.

Our study has several limitations. We based our analysis on the full dataset to have maximum statistical power instead of leaving a part of the dataset out for validation. While our analysis only included a single isolate per participant (i.e., a single time-point), samples from multiple time-points could have provided further information about time-dependent relationships between identified associations, viral genome diversification, and markers of disease progression. Additionally, the inclusion of host genetics in the association analysis between Shannon entropy and VL could have furthered our understanding of the relationship between the intrahost diversity at the specific positions and host immunotypes. Furthermore, success of sequencing varied across both samples and genomic positions which limited the number of samples and genomic positions that could be analysed, and we only included variants presented by at least 1% of covering sequencing reads which may make us disregard variation. Moreover, sequencing was based on two amplicons covering the majority but not the full length of the HIV genome which prevented us from identifying associations outside of the sequenced amplicons. Sequence reads were aligned against the HXB2 reference genome which belongs to subtype B and, therefore, some reads from other subtypes might not have been aligned at the most variable genomic regions. Finally, no participants with low VL were included in the analysis because of the nature of the cohort and, therefore, our analysis fails to identify associations dependent on inclusion of such participants.

In conclusion, by using a large and demographically diverse cohort of HIV-infected treatment-naive participants, we confirmed that the highest intrahost diversity is accumulated in the Env region. Furthermore, we showed that both sample's mean intrahost genetic diversity and the intrahost genetic diversity at the 16 HIV genomic positions have a positive relationship with VL, and that intrahost genetic diversity is a better marker of disease progression than HIV variants identified from consensus sequences. Accordingly, HIV intrahost genetic diversity should be taken into account when analysing HIV genomes and the clinical outcomes.

## Supporting information

**S1 Table. Characteristics of 2,650 participants from the START trial included in this analysis.** IQR (interquartile range).
(DOCX)

**S2 Table. HIV genomic positions for which ≤800 samples had a minimum of 500-fold genome sequencing depth at position.**
(DOCX)

**S3 Table. HIV genome's mean Shannon entropy at the most variable positions.**
(DOCX)

**S4 Table. Summary of the total considered positions, explained variance and Pearson's correlation coefficient based on the infection duration in the robustness analysis of the results when changing the sequencing depth requirement for positions to be included from 500-fold to 300-fold and 1,000-fold, respectively.**
(DOCX)

**S5 Table. Linear regression analysis of predictors of the viral load (VL).**
(DOCX)

**S6 Table. Joint multiple linear regression model with the Shannon entropy values from significantly associated AA positions.** P-values below the 0.05 threshold are marked in bold.
(DOCX)

**S7 Table. Average RMSE values for the train and test datasets from the 5-fold cross validation analysis for the 16 Shannon entropy HIV AA positions associated with VL.**
(DOCX)

**S8 Table. Joint model of multiple linear regression with the consensus AA variants which were significantly associated with viral load. P-values below the 0.05 threshold are marked in bold.**
(DOCX)

**S1 Fig. An illustration of the HIV genome and the positions covered by the two amplicons sequenced in this study.**
(DOCX)

**S2 Fig. The density plots of the mean Shannon entropy in different races of the 36 most variable positions.**
(DOCX)

**S3 Fig. The density plots of the mean Shannon AA entropy in different subtypes of the 36 most variable positions.**
(DOCX)

**S4 Fig. Relationship between log10 viral load and Shannon entropy in the 16 most significantly associated HIV amino acid positions. Blue line marks smoothed conditional means.**
(DOCX)

**S5 Fig. Pairwise correlation matrix between covariates (bottom triangle).**
(DOCX)

## Acknowledgments

We would like to thank all participants in the START trial and all trial investigators (see N Engl J Med 2015; 373:795–807 for the complete list of START investigators). The START trial was supported by the National Institute of Allergy and Infectious Diseases, Division of AIDS (United States) (NIH Grants UM1-AI068641, UM1-AI120197 and 1U01-AI36780), National Institutes of Health Clinical Center, National Cancer Institute, National Heart, Lung, and Blood Institute, Eunice Kennedy Shriver National Institute of Child Health and Human Development, National Institute of Mental Health, National Institute of Neurological Disorders and Stroke, National Institute of Arthritis and Musculoskeletal and Skin Diseases, Agence Nationale de Recherches sur le SIDA et les Hépatites Virales (France), National Health and Medical Research Council (Australia), National Research Foundation (Denmark), Bundes ministerium für Bildung und Forschung (Germany), European AIDS Treatment Network, Medical Research Council (United Kingdom), National Institute for Health Research, National Health Service (United Kingdom), and University of Minnesota. Antiretroviral drugs were donated to the central drug repository by AbbVie, Bristol-Myers Squibb, Gilead Sciences, GlaxoSmithKline/ViiV Healthcare, Janssen Scientific Affairs, and Merck.

## Author Contributions

**Conceptualization:** Jens D. Lundgren, Rasmus L. Marvig.

**Formal analysis:** Migle Gabrielaite, Marc Bennedbæk, Malthe Sebro Rasmussen.

**Methodology:** Migle Gabrielaite, Marc Bennedbæk, Malthe Sebro Rasmussen, Jens D. Lundgren, Rasmus L. Marvig.

**Supervision:** Jens D. Lundgren, Rasmus L. Marvig.

**Visualization:** Migle Gabrielaite.

**Writing – original draft:** Migle Gabrielaite, Marc Bennedbæk.

**Writing – review & editing:** Migle Gabrielaite, Marc Bennedbæk, Malthe Sebro Rasmussen, Virginia Kan, Hansjakob Furrer, Robert Flisiak, Marcelo Losso, Jens D. Lundgren, Rasmus L. Marvig.

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
