## [Decision Letter · Decision Letter 0]

13 Jan 2022

Dear Ms. Gabrielaite,

Thank you very much for submitting your manuscript "Deep-sequencing of viral genomes from treatment-naive HIV-infected persons shows positive association between intrahost genetic diversity and viral load" for consideration at PLOS Computational Biology.

As with all papers reviewed by the journal, your manuscript was reviewed by members of the editorial board and by several independent reviewers. In light of the reviews (below this email), we would like to invite the resubmission of a significantly-revised version that takes into account the reviewers' comments.

We cannot make any decision about publication until we have seen the revised manuscript and your response to the reviewers' comments. Your revised manuscript is also likely to be sent to reviewers for further evaluation.

Sincerely,

Nicola Segata

Associate Editor

PLOS Computational Biology

Rob De Boer

Deputy Editor

PLOS Computational Biology

Reviewer's Responses to Questions

**Comments to the Authors:**

Reviewer #1: Gabirelaite et al. report the results from a large cross-sectional study of a HAART-naïve HIV-1 infected cohort aimed at exploring the correlation between viral intra-host genetic diversity, analyzed by deep sequencing, and viral load, which can be considered a marker of disease progression.

Given the remarkable size and the demographic heterogeneity of the sample analyzed, this study is quite an unprecedented effort, which includes more than 2000 subjects. The scientific research question, i.e. whether the intra-host sequence diversity found in a chronically infected individual correlates with set point viral load, is very interesting and relevant both for the potential implication for predicting clinical outcome and for understanding the dynamics and the mechanism of virus evolution in vivo. Results presented show a weak, though significant, correlation with diversity, assessed by calculating Shannon Entropy. According to the study, HIV-1 env is the genetic region with the highest concentration of highly diverse position, in agreement with the high selective pressure to which Env is subjected.

Overall, I think the outcome improves our knowledge of intra-host viral evolution, contributing to the notion that intra-host diversity somehow benefits the persistence of virus replication.

I wouls like to suggest a few clarifications, for the benefit of the reader:

1) What is the variability of sequencing depth of the samples that remained after excluding from analyses those with 500 or less codon positions covered by ≥500 reads? Was there an objective criterion for choosing a 500 reads threshold?

2) On a similar topic, what were the criteria for excluding positions for which ≤800 samples had mean Shannon entropy calculations? How was the 800 samples threshold decided?

3) given the highest concentration of positions featuring high level of diversity within Env, it would be useful to include a short comment on the location of those positions relativel to the most prominent features of Env, such as the variable loops, which could be relevant to immune escape from humoral immunity.

Reviewer #2: This paper describes the relationship between nucelotide diversity of HIV, as measured by Shannon entropy, and viral load in subjects who are treatment naiive to HIV. No major justification is given as to why this is an interesting association to study. The manuscript also discusses but does not focus on the relationship between Shannon entropy and disease progression, but perhaps underplays this positive result as it has already been described.

Major concerns:

-The implications of finding a correlation of entropy with viral load are not clear. Perhaps if this association could explain disease progression by combination with some other data from the patients, that would be a major advance. Alternatively, the authors might focus on the uniqueness of this cohort, and their lack of treatment. Are certain loci more or less likely to be diverse than when compared to published data sets from treated individuals?-

-The authors show that diversity 'explains' VL, but the directionality of this association is not clear-- could higher VL enable more diversity?

-A separate test set is needed to provide confidence in model predictions (association of single AA positions and predictive ability of entropy on VL).

-It would be nice to see graphs of VL vs entropy for at least of few of the associated positions. Are a few extreme points driving these associations?

-Is VL associated with entropy at the gene level?

**Have the authors made all data and (if applicable) computational code underlying the findings in their manuscript fully available?**

Reviewer #1: Yes

Reviewer #2: None

PLOS authors have the option to publish the peer review history of their article (what does this mean?). If published, this will include your full peer review and any attached files.

Reviewer #1: No

Reviewer #2: No
---

## [Decision Letter · Decision Letter 1]

4 May 2022

Dear Ms. Gabrielaite,

Thank you very much for submitting your manuscript "Deep-sequencing of viral genomes from treatment-naive HIV-infected persons shows positive association between intrahost genetic diversity and viral load" for consideration at PLOS Computational Biology. As with all papers reviewed by the journal, your manuscript was reviewed by members of the editorial board and by several independent reviewers. The reviewers appreciated the attention to an important topic. Based on the reviews, we are likely to accept this manuscript for publication, provided that you modify the manuscript according to the review recommendations.

When you are ready to resubmit, please upload the following

Sincerely,

Nicola Segata

Associate Editor

PLOS Computational Biology

Rob De Boer

Deputy Editor

PLOS Computational Biology

[LINK]

Reviewer's Responses to Questions

**Comments to the Authors:**

Reviewer #1: I am satisfied with authors, answers and modifications in the revised manuscript.

Reviewer #2: I found some responses to my previous concerns to be unsatisfactory:

1) I understand that the authors do not want to change the scope of their manuscript, and that is certainly a reasonable choice. However, given the broad audience of this journal, it would be helpful to better explain to the reader why a relationship between VL and diversity are interesting in the abstract. It seems to me from the discussion that the authors do not believe that diversity might drive VL. Might the identified relationship allow a future researcher to model other predictors of VL by using diversity as a proxy? Is there some other utility? If not, the authors should emphasize in the abstract what they mention in the introduction – previous studies have found conflicting relationships between VL and intrahost diversity. [Minor concern]

2) The authors have responded to my question about specific genes being associated with VL by presenting what appears to be a fundamentally different analysis than performed for the genome as a whole (simple correlation vs multivariate regression). While a single simple correlation was done for sample mean entropy vs VL, what is reported is a correlation coefficient (and no p value), whereas only p-values (and not correlation coefficients) are reported for the gene-level analyses. [Minor concern]

3) Given the centrality of the machine learning model to the data set, and the large number of positions tested, overfitting is a major possibility, particularly for the report of specific loci. I think it is appropriate and necessary for the authors to repeat their model using 800 samples as a training set and reserving 200 as a test set. The authors can compare RMSE in training and test sets and see if the same 16 positions are most correlated with VL in the test set. [Major criticism]

I also have some other concerns not mentioned in the first review:

1) The abstracts says that “genetic diversity … better disease progression marker than variation of HIV consensus sequences.” A similar claim is repeated twice in the discussion. However, I was unable to locate where this comparison emerges from, as I do not see a model presented that attempts to predict VL from consensus sequence. [Major criticism]

2) On closer reading, I now see that the manuscript says “ In total, the final regression model (i.e., sample’s mean Shannon entropy *together with all the covariates*) explained 10.3% of the variance in VL.” How much variance in VL does a model that includes the covariates (including duration of infection!) but not mean Shannon entropy explain? In other words, how much of the performance is due to the Shannon entropy parameter? [Major criticism]

3) On page 8—“It should be noted that the samples of the excluded genomes had a significantly lower VL than the included HIV genomes”. Did coverage within the included samples also correlate with VL? This would suggest residual confounding.

I also found the answer to reviewer #1s question unsatisfactory:

1) Given the per-base coverage threshold of 500 reads, it is more appropriate to exclude any samples with less than this coverage averaged across their genome.

Reviewer #3: In this work, the authors study the correlation between intra-host genetic diversity during HIV-1 infections and viral load which, in turn, is a marker of disease progression. The issue is interesting from several standpoints going from a better understanding of the rules governing virus-host interactions to improving the precision of predicting clinical outcomes of the infection. The take home message of the work improves knowledge in these fields.

I have a few minor issues that the authors could address, in my opinion, to improve the manuscript.

1. In the background section of the abstract they state that "viral intrahost genetic diversity remains a major obstacle to the eradication of HIV". I think that the word "eradication" is evocative of making the virus disappear from the human population. If so, I do not see a direct relationship between intra-host genetic diversity and eradication. Inter-host diversity is rather the important issue. Intra-host diversity is rather correlated to "eradication" of the virus from the patient, case for which I rather find more appropriate to use the term "cure". The same comment applies to the first line of the author summary.

2. Page 9, line 6 from bottom. How the conclusion that Shannon entropy accounts for 4.14% of the VL variance was reached is not straightforward (at least for me). For the sake of clarity, it should be indicated how this value (and, similarly, those indicated at lines 3 and 4 of page 10) was deduced.

3. Also, at line 4 from the top of page 10, it is referred the effect of the interaction between Shannon entropy and duration of infection. Where is this parameter measured?

4. Last sentence of the Results section. It is an interesting observation. Wouldn't it be worth to make a short comment on this in the discussion section?

5. Lines 5 to 4 from bottom of page 12. It could be interesting to add a comment for the fact that the study by Bello and coll. concerned long-term non-progressors while the present one did not (even if in that case the directionality of the correlation could be determined while in this work it could not).

**Have the authors made all data and (if applicable) computational code underlying the findings in their manuscript fully available?**

Reviewer #1: Yes

Reviewer #2: **No: **I do not see a code availability statement. A supplementary table of entropy x sample would also be helpful.

Reviewer #3: Yes

PLOS authors have the option to publish the peer review history of their article (what does this mean?). If published, this will include your full peer review and any attached files.

Reviewer #1: No

Reviewer #2: No

Reviewer #3: **Yes: **Matteo Negroni

Figure Files:

Data Requirements:

Reproducibility:

References:

---

## [Decision Letter · Decision Letter 2]

18 Jul 2022

Dear Ms. Gabrielaite,

Thank you very much for submitting your manuscript "Deep-sequencing of viral genomes from treatment-naive HIV-infected persons shows positive association between intrahost genetic diversity and viral load" for consideration at PLOS Computational Biology.

As with all papers reviewed by the journal, your manuscript was reviewed by members of the editorial board and by several independent reviewers. In light of the reviews (below this email), we would like to invite the resubmission of a significantly-revised version that takes into account the reviewers' comments.

We cannot make any decision about publication until we have seen the revised manuscript and your response to the reviewers' comments. Your revised manuscript is also likely to be sent to reviewers for further evaluation.

Sincerely,

Nicola Segata

Associate Editor

PLOS Computational Biology

Rob De Boer

Deputy Editor

PLOS Computational Biology

Reviewer's Responses to Questions

**Comments to the Authors:**

Reviewer #2: The revised manuscript reports an association between viral load and entropy in cross-sectional HIV samples. Previous papers have found conflicting results between viral load and entropy, and this paper attempts to resolve this issue.

However, I have not been convinced that coverage and other artifacts have been appropriately accounted for. The authors’ apparent unwillingness to rerun the analysis with a held-out test set, or with any change of parameters (e.g. coverage threshold for sample inclusion), does not invoke confidence. If the authors are not willing to demonstrate rigor in this analysis, the impact of this finding seems low in the context of previous studies showing conflicting results.

In addition, the abstract concludes that entropy “could be used as a better disease progression marker than HIV consensus sequence variants, especially in infections of longer duration” by pointing to a previous publication. Unless the analysis is performed on the same exact sample set (see below for one possible suggestion), this comparison should not be included prominently in the abstract.

Specific points:

-The authors have demonstrated that coverage correlates with VL in the data set. Yet, to my knowledge, no attempt has been made to demonstrate that coverage doesn’t impact entropy and/or correlations of entropy and VL. One way to show this would be to repeat the ANOVA analysis or linear regression in Supplementary Table 4 with coverage as a variable (and also include a model that just includes coverage for comparison with the model that just includes entropy). Another would be to remove samples with low coverage (e.g. < 500 or < 750 fold coverage) and repeat the analysis; as it stands, samples with average coverage below the per-position threshold for calculation of entropy are included, which is not best practice.

-If the authors want to include a comparison of entropy vs consensus sequence prediction of viral load, an analysis of just the PCs needs to be included in Supplementary Table 4. Reference to a different but related analysis is not a sufficient replacement.

-Page 11: It is not exactly clear what is meant by “the joint model including all 16 positions”. Does this also include PCs? One possible explanation for some of the variants being unpredictive in this context is that some positions are more likely to have diversity due to alignment errors, which are more likely to occur on particular genomic backgrounds.

Writing issues and minor suggestions:

-Page 9: What is the nature of positions in the env gene that do not have high coverage in a lot of samples? Are these impact by indels or similar?

-Page 9: Why is the % of positions in each gene an important number to discuss in the results?

-Page 10: In the results, PCs come out of seemingly nowhere. While this is mentioned in the results, the authors should walk the reader through what these are used for (genetic background of virus).

-Page 10: “Bonferroni corrected p-value threshold” would be more straightforward than “our p-value threshold”

-Page 11: The authors should speculate on why the by-gene entropy analysis fails to associate with VL when the average of these values does associate with VL (unless I am misunderstanding what was done).

-Page 11: It is not exactly clear what is meant by “the joint model including all 16 positions”. Does this also include PCs?

-Page 12: Final sentence. What evidence is being used to support this important claim?

Reviewer #3: The authors have nicely followed the suggestions made. I am happy with this revised version.

Reviewer #4: This paper examines the relationship between intrahost viral diversity and viral load in patients enrolled in a clinical trial of early ART initiation. The study design of the parent trial means that participants tended to be either relatively early in their infection, or more likely to be relatively slow progressors, having maintained a CD4 count in the normal range (>500) without treatment. On the other hand, viral controllers are excluded from this analysis by virtue of not having sufficiently high viral load for amplification and sequencing (cut-off 1000 copies/ml).

Although previous studies have suggested that the extent of intrahost viral diversity is likely to vary over the course of HIV infection (for example, being lower in late disease when host immune responses are less effective), this is a cross-sectional rather than a longitudinal study, with only a single sample being studied in each patient. I think there are some concerns about trying to relate this to viral load in patients early in infection, during the very dynamic stages of primary HIV infection - in many ways I would be more comfortable if the analysis focused on patients who had already reached set-point viral load (usually some time between 6 and 24 months), although I appreciate that the authors have calculated the mean Shannon entropy separately for patients infected less than 6 months, 6-24 months and >24 months. I think it is also important to note that the amplicons do not cover the full length of the HIV genome, missing out p17 as well as vif and vpu.

I am not convinced that the debate about the relationship between viral diversity and viral load in HIV infection is as intense as the authors suggest. If this is a key argument about the novelty of this paper, then I think the authors should have made more effort to review the literature about HIV diversity and viral load/clinical outcome, focusing on the major studies on this topic. A number of key papers, which were not cited, have come to the same conclusions as the authors, for example: Esbjornsson et al, NEJM, 2012 (which is a longitudinal study that demonstrates increasing diversity over time up to a “diversity threshold” heralding the onset of AIDS), and a very similar study to this one, by Blanquart et al in PLOS Biology 2017, in which the authors focus specifically on the early stages of HIV infection and conclude that viral diversity contributes around one third of set-point viral load variability.

In contrast, the main argument to the contrary was reported by Wolinsky et al (Science 1996) in a select group of patients with very rapid disease progression, who were assumed to have mounted very limited immune responses to the virus.

I don't think anyone in the HIV field will be surprised to hear that the greatest diversity is found in the env gene, so the main novelty in the paper is the identification of amino acids where variation correlates significantly with viral load: several of these are found in the vicinity of T-cell epitopes but the relationship between env amino acids and antibody-binding epitopes has not been considered - I think this would add considerably to the paper (see, for example, Andrews et al, Scientific Reports, 2018).

**Have the authors made all data and (if applicable) computational code underlying the findings in their manuscript fully available?**

Reviewer #2: **No: **A promise is made to upload these items, but this has not yet been delivered on.

Reviewer #3: None

Reviewer #4: Yes

PLOS authors have the option to publish the peer review history of their article (what does this mean?). If published, this will include your full peer review and any attached files.

Reviewer #2: No

Reviewer #3: **Yes: **Matteo Negroni

Reviewer #4: No
---

## [Decision Letter · Decision Letter 3]

10 Oct 2022

Dear Ms. Gabrielaite,

Thank you very much for submitting your manuscript "Deep-sequencing of viral genomes from treatment-naive HIV-infected persons shows positive association between intrahost genetic diversity and viral load" for consideration at PLOS Computational Biology. As with all papers reviewed by the journal, your manuscript was reviewed by members of the editorial board and by several independent reviewers. The reviewers appreciated the attention to an important topic. Based on the reviews, we are likely to accept this manuscript for publication, providing that you modify the manuscript according to the review recommendations.

Sincerely,

Nicola Segata

Academic Editor

PLOS Computational Biology

Rob De Boer

Section Editor

PLOS Computational Biology

Reviewer's Responses to Questions

**Comments to the Authors:**

Reviewer #2: The manuscript has been much improved in terms of rigor by the addition of new analyses. While cross-validation is not the same as an independent test set, the direct analyses of the effects of coverage and sequence validation has satisfied my concerns.

I do not want to unnecessarily burden the authors, but one thing the authors and editor might consider together is improving the readability of the manuscript by including a figure in the main text that summarizes the models tried and the relative predictive power of these models. Having to go to the supplement to understand the evidence for key claims (in particular the relative predictive power of sequence/PC and entropy, mentioned in abstract) is not ideal.

Reviewer #4: The authors have addressed only one of my comments, about the influence of time since infection on the analysis, and have either ignored or failed to address the other comments. I think that the question of the novelty of their findings remains to be answered.

**Have the authors made all data and (if applicable) computational code underlying the findings in their manuscript fully available?**

Reviewer #2: Yes

Reviewer #4: None

PLOS authors have the option to publish the peer review history of their article (what does this mean?). If published, this will include your full peer review and any attached files.

Reviewer #2: No

Reviewer #4: No

Figure Files:

Data Requirements:

Reproducibility:

References:

---

## [Decision Letter · Decision Letter 4]

17 Nov 2022

Dear Ms. Gabrielaite,

Thank you very much for submitting your manuscript "Deep-sequencing of viral genomes from a large and diverse cohort of treatment-naive HIV-infected persons shows associations between intrahost genetic diversity and viral load" for consideration at PLOS Computational Biology. As with all papers reviewed by the journal, your manuscript was reviewed by members of the editorial board and by several independent reviewers. The reviewers appreciated the attention to an important topic. Based on the reviews, we are likely to accept this manuscript for publication, providing that you modify the manuscript according to the review recommendations.

Sincerely,

Nicola Segata

Academic Editor

PLOS Computational Biology

Rob De Boer

Section Editor

PLOS Computational Biology

Reviewer's Responses to Questions

**Comments to the Authors:**

Reviewer #2: I appreciate that the authors attempted to meet my request for a Table 2 for a direct comparison of models. However, this table is confusing and not helpful in its current form-- as it is hard to compare prose across the various models. Information on entropy should be removed (this doesn't need to summarize the entire paper!) and versions of the same exact model should be compared in a simple table, where each row corresponds to the data included in that model and the value presented in each cell shows the total performance of the model (e.g. % of VL variance explained). Values of table should only include numbers, and explanations of what these numbers mean should be included in the headers of the column(s) (indicating that the equivalent metrics are being compared across models).

Alternatively, the authors could remove this table.

Reviewer #4: I am glad that the authors have now addressed my comments.

**Have the authors made all data and (if applicable) computational code underlying the findings in their manuscript fully available?**

Reviewer #2: None

Reviewer #4: Yes

PLOS authors have the option to publish the peer review history of their article (what does this mean?). If published, this will include your full peer review and any attached files.

Reviewer #2: No

Reviewer #4: No

Figure Files:

Data Requirements:

Reproducibility:

References:

---

## [Editor Report · Decision Letter 5]

23 Nov 2022

Dear Ms. Gabrielaite,

We are pleased to inform you that your manuscript 'Deep-sequencing of viral genomes from a large and diverse cohort of treatment-naive HIV-infected persons shows associations between intrahost genetic diversity and viral load' has been provisionally accepted for publication in PLOS Computational Biology.

Best regards,

Nicola Segata

Academic Editor

PLOS Computational Biology

Rob De Boer

Section Editor

PLOS Computational Biology

---

## [Editor Report · Acceptance letter]

23 Dec 2022

PCOMPBIOL-D-21-02241R5 

Deep-sequencing of viral genomes from a large and diverse cohort of treatment-naive HIV-infected persons shows associations between intrahost genetic diversity and viral load

Dear Dr Gabrielaite,

I am pleased to inform you that your manuscript has been formally accepted for publication in PLOS Computational Biology. Your manuscript is now with our production department and you will be notified of the publication date in due course.

With kind regards,

Zsofi Zombor
